# Whole-chromosome hitchhiking driven by a male-killing endosymbiont

**Simon H. Martin**[1,2]\*, **Kumar Saurabh Singh**[3], **Ian J. Gordon**[4], **Kennedy Saitoti Omufwoko**[5,6], **Steve Collins**[7], **Ian A. Warren**[2], **Hannah Munby**[2], **Oskar Brattström**[2], **Walther Traut**[8], **Dino J. Martins**[5,6], **David A. S. Smith**[9], **Chris D. Jiggins**[2], **Chris Bass**[3], **Richard H. ffrench-Constant**[3]

**1** Institute of Evolutionary Biology, University of Edinburgh, Edinburgh, United Kingdom, **2** Department of Zoology, University of Cambridge, Cambridge, United Kingdom, **3** Centre for Ecology and Conservation, University of Exeter, Penryn Campus, Penryn, United Kingdom, **4** Center of Excellence in Biodiversity and Natural Resource Management, University of Rwanda, Huye, Rwanda, **5** Department of Ecology and Evolutionary Biology, Princeton University, Princeton, United States of America, **6** Mpala Research Centre, Nanyuki, Kenya, **7** African Butterfly Research Institute, Nairobi, Kenya, **8** Institut für Biologie, Universität Lübeck, Lübeck, Germany, **9** Natural History Museum, Eton College, Windsor, United Kingdom

\* simon.martin@ed.ac.uk

**Data Availability Statement:** Raw genomic data and assemblies are available from GenBank (project accession numbers PRJNA448181 and PRJEB35880, and individual sample accessions are provided in S9 Table). All processed data files

## Abstract

Neo-sex chromosomes are found in many taxa, but the forces driving their emergence and spread are poorly understood. The female-specific neo-W chromosome of the African monarch (or queen) butterfly *Danaus chrysippus* presents an intriguing case study because it is restricted to a single 'contact zone' population, involves a putative colour patterning supergene, and co-occurs with infection by the male-killing endosymbiont *Spiroplasma*. We investigated the origin and evolution of this system using whole genome sequencing. We first identify the 'BC supergene', a broad region of suppressed recombination across nearly half a chromosome, which links two colour patterning loci. Association analysis suggests that the genes *yellow* and *arrow* in this region control the forewing colour pattern differences between *D. chrysippus* subspecies. We then show that the same chromosome has recently formed a neo-W that has spread through the contact zone within approximately 2,200 years. We also assembled the genome of the male-killing *Spiroplasma*, and find that it shows perfect genealogical congruence with the neo-W, suggesting that the neo-W has hitchhiked to high frequency as the male-killer has spread through the population. The complete absence of female crossing-over in the Lepidoptera causes whole-chromosome hitchhiking of a single neo-W haplotype, carrying a single allele of the BC supergene and dragging multiple non-synonymous mutations to high frequency. This has created a population of infected females that all carry the same recessive colour patterning allele, making the phenotypes of each successive generation highly dependent on uninfected male immigrants. Our findings show how hitchhiking can occur between the physically unlinked genomes of host and endosymbiont, with dramatic consequences.

underlying all figures are available from the Dryad digital repository: https://doi.org/10.5061/dryad.9kd51c5d0. Scripts used for data analysis are available from https://github.com/simonhmartin/genomics_general.

**Funding:** This work was funded by European Research Council (https://erc.europa.eu) European Union Horizon 2020 research and innovation programme grant 646625 (CB), ERC grant 339873 (CDJ), National Geographic Society (https://www.nationalgeographic.org) Research Grant WW-138R-17 (IJG), and a Royal Society (https://royalsociety.org) University Research Fellowship URF\R1\180682 (SHM). The funders had no role in study design, data collection and analysis, decision to publish, or preparation of the manuscript.

**Competing interests:** The authors have declared that no competing interests exist.

**Abbreviations:** chr15, Chromosome 15; COI, Cytochrome Oxidase Subunit I; GDP, glycerophosphoryl diester phosphodiesterase; LD, linkage disequilibrium; Mb, megabase.

# Introduction

Structural changes to the genome play an important role in evolution by altering the extent of recombination among loci. This is best studied in the context of chromosomal inversions that cause localised recombination suppression, and can be favoured by selection if they help to maintain clusters of co-adapted alleles (or 'supergenes') in the face of genetic mixing [1–4]. A greater extent of recombination suppression occurs in the formation of heteromorphic sex chromosomes, which can link sex-specific alleles similarly to supergenes [5]. However, suppressed recombination can also have costs. In particular, male-specific Y and female-specific W chromosomes can be entirely devoid of recombination, making them vulnerable to genetic hitchhiking and the accumulation of deleterious mutations through 'Muller's ratchet', which may explain their deterioration over time [6–8]. These contrasting benefits and costs of recombination suppression are of particular interest in the evolution of neo-sex chromosomes, which can form through fusion of autosomes to existing sex chromosomes. There is accumulating evidence that neo-sex chromosomes are common in animals [9–15], but the processes underlying their emergence, spread, and subsequent evolution have not been widely studied. In particular, there are few studied examples of recently formed neo-sex chromosomes that are not yet fixed in a species.

The African monarch (or queen) butterfly *Danaus chrysippus* provides a unique test case for the causes and consequences of changes in genome architecture and recombination suppression. Like its American cousin (*D. plexippus*), it feeds on milkweeds and has bright colour patterns that warn predators of its distastefulness. However, within Africa, *D. chrysippus* is divided into four subspecies with distinct colour patterns and largely distinct ranges (Fig 1A). Predator learning should favour the maintenance of a single monomorphic warning in any single area. For this reason, researchers have long been puzzled by the large polymorphic contact zone in East and Central Africa, where all four *D. chrysippus* subspecies meet and interbreed [16–18] (Fig 1A). Crosses have shown that colour pattern differences between the subspecies are controlled by Mendelian autosomal loci, including the tightly linked 'B' and 'C' loci (putatively a 'BC supergene' [19]) that define three common forewing patterns [20,21] (Fig 1A). However, crosses with females from the contact zone revealed that the BC chromosome has become sex linked, forming a neo-W that is unique to this population [19,22]. Because female meiosis is achiasmatic (it lacks crossing-over) in the Lepidoptera, the formation of a neo-W would instantaneously cause perfect linkage, not just of the B and C loci but of an entire non-recombining chromosome, along with other maternally inherited DNA.

What is particularly striking is that the presence of the neo-W coincides with infection by a maternally inherited 'male-killer' endosymbiont related to *Spiroplasma ixodetis*, which kills male offspring and leads to highly female-biased sex ratios where infection is common [22–24]. The combination of neo-W and male-killing is expected to dramatically alter the inheritance and evolution of the BC chromosome [22,25]: Infected females typically give rise to all-female broods who should always inherit the same colour patterning allele on their neo-W, along with the male-killer, while the other maternal allele is systematically eliminated in the dead sons (Fig 1B), forming a genetic sink for all colour pattern alleles not on the neo-W. It has been suggested that the restriction of male-killing to females with the neo-W, and only in the region in which hybridisation occurs between subspecies, may not be a coincidence [19,22,25–27]. However, the genomic underpinnings of this system—the genetic controllers of colour pattern, the source and spread of the neo-W, and its relationship with the male-killer—have until now remained a mystery. We generated a reference genome for *D. chrysippus* and used whole genome sequencing of population samples to uncover the interconnected

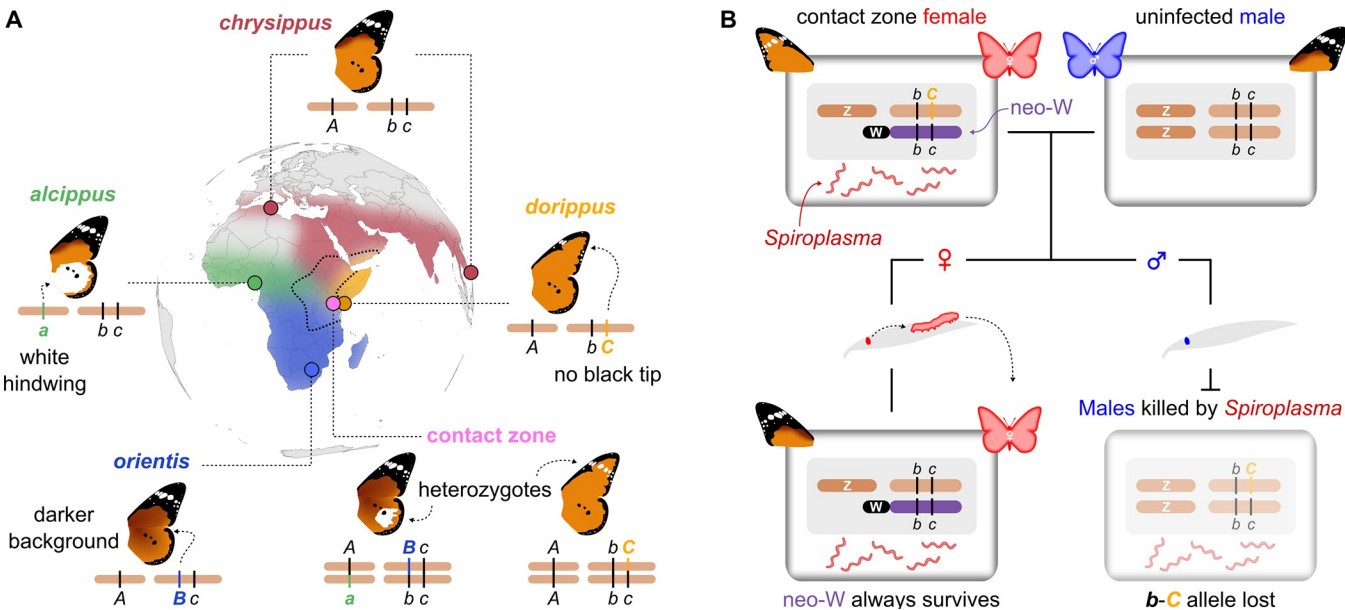

**Fig 1. Geography and genetics of colour pattern.** (**A**) Approximate ranges of the four subspecies of *D. chrysippus*, with the contact zone outlined. Sampling locations for each of the subspecies and the contact zone are indicated. Cartoon chromosomes show the genotypes of each subspecies at the A (white hindwing patch), B (brown background colour), and C (forewing tip) colour patterning loci, based on previous crosses [20]. Note the linkage of B and C, putatively forming a 'BC supergene' [19]. Two examples of heterozygotes that can be found in the contact zone are shown. Note that *Cc* heterozygotes can exhibit the transiens phenotype with white markings on the forewing, with approximately 50% penetrance. (**B**) Model showing how fusion of the BC autosome to the W chromosome has produced a neo-W (purple) in contact zone females (top left), while males have two autosomal copies of the BC chromosome (top right). Daughters inherit the neo-W, while sons inherit the other BC chromosome haplotype from their mother. The latter allele is then lost due to male-killing by *Spiroplasma*.

evolution of the BC supergene, neo-W, and *Spiroplasma*. Our findings reveal a recent whole-chromosome selective sweep caused by hitchhiking between the host and endosymbiont genomes.

## Results and discussion

### Identification of the BC supergene

We assembled a high-quality draft genome for *D. chrysippus*, with a total length of 322 mega-bases (Mb), a scaffold N50 length of 0.63 Mb, and a BUSCO [28] completeness score of 94% (S1–S8 Tables). We then further scaffolded the genome into a pseudo-chromosomal assembly based on homology with the *Heliconius melpomene* genome [29–31], accounting for known fusions that differentiate these species [9,30,32] (S1 Fig). We also resequenced 42 individuals representing monomorphic populations of each of the four subspecies and a polymorphic population from a known male-killing hotspot near Nairobi, in the contact zone (Fig 1A, S9 Table).

To identify the putative BC supergene, we scanned for genomic regions showing high differentiation between the subspecies and an association with colour pattern. Genetic differentiation ($F_{ST}$) and excessive divergence ($d_{XY}$) is largely restricted to a handful of broad peaks, with a background $F_{ST}$ of approximately zero (Fig 2A, S2 Fig, and S3 Fig). This low background level implies a nearly panmictic population across the continent. The effective population size appears to be very large, as average genome-wide diversity at putatively neutral 4-fold degenerate third codon positions is 0.042, which is among the highest values reported for animals [33,34]. The islands of differentiation that stand out from this background imply selection

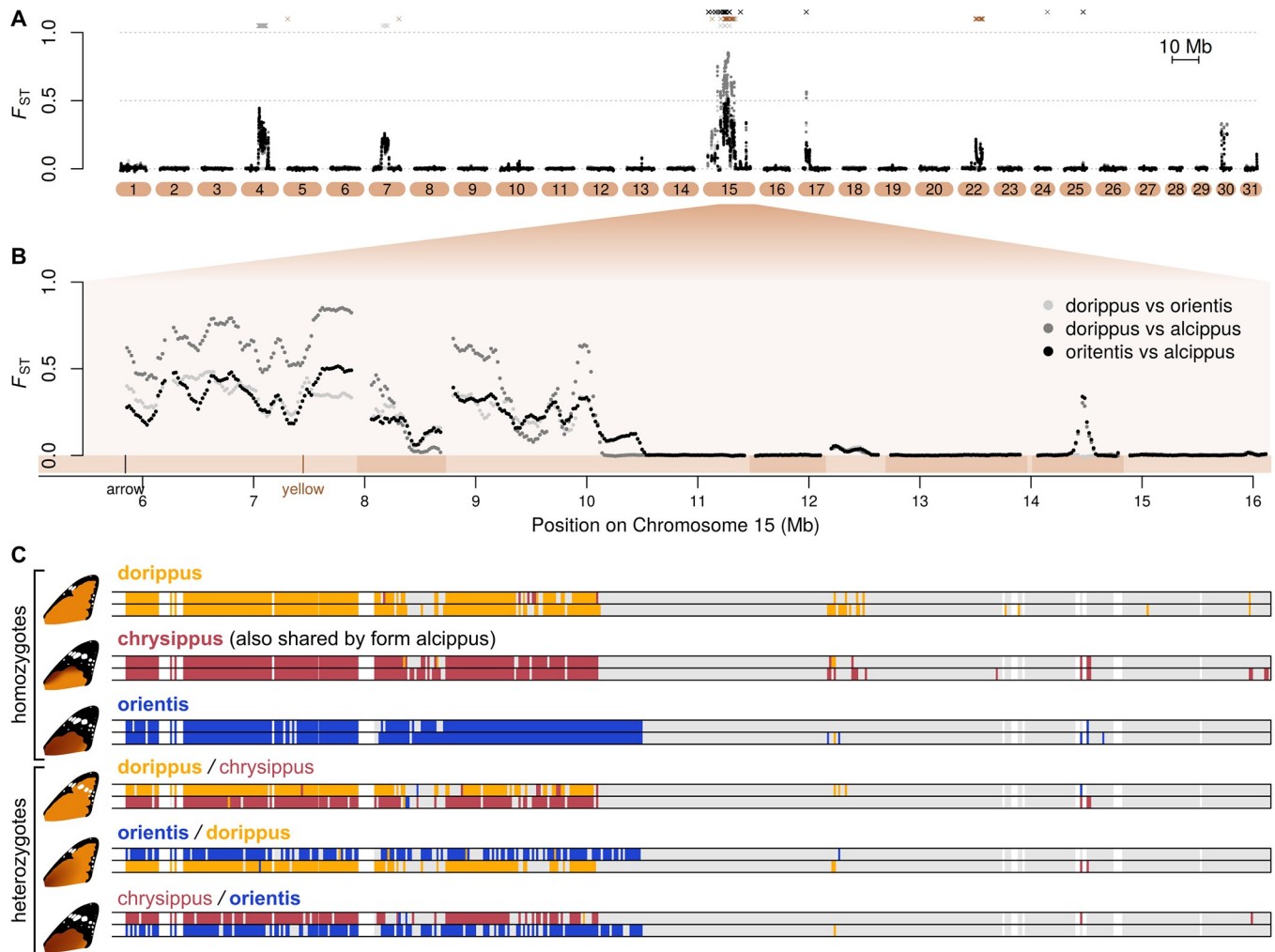

**Fig 2. Identification of the BC supergene on Chromosome 15.** (**A**) Pairwise genetic differentiation ($F_{ST}$), plotted in 100-kb sliding windows with a step size of 20 kb across all chromosomes. Three subspecies pairs with sample sizes ≥6 are shown (see legend in panel B), revealing strong overlap in patterns of differentiation. Locations of SNPs most strongly associated with the A, B, and C loci (Wald test, 99.99% quantile) are plotted above in grey, brown, and black, respectively. See S2 Fig for a more detailed plot. (**B**) Expanded plot of $F_{ST}$ across a 10-Mb portion of Chromosome 15 (chr15). Note that the first approximately 6 Mb of the chromosome is not included in this plot due to complex structural variation (see main text). Scaffolds are indicated below the plot in alternating shades. The locations of our most likely candidate genes for B (*yellow*) and C (*arrow*) are indicated. (**C**) Allelic clustering on chr15 in six representative individuals: three homozygotes and three heterozygotes (see S4 Fig for all individuals, and see panel B for chromosome positions). Coloured blocks indicate 20-kb windows, in which sequence haplotypes could be assigned to one of three clusters based on pairwise genetic distances (see Methods for details). Windows in grey indicate insufficient relative divergence to be assigned to a cluster, and white indicates missing data. Alleles are named according to the form in which they occur. In heterozygotes, the name of the dominant allele is in bold. Data deposited in the Dryad repository [36].

for local adaptation maintaining particular differences between the subspecies, similar to patterns seen between geographic races of *Heliconius* butterflies [35]. However, here the peaks of differentiation are broad, covering several Mb, implying some mechanism of recombination suppression such as inversions that differentiate the subspecies.

The inclusion of the polymorphic contact-zone samples, and the fact that three of the subspecies each carry a unique colour pattern allele (Fig 1A), allowed us to identify particular differentiated regions associated with the three major colour pattern traits. A region of approximately 3 Mb on Chromosome 4 is associated with the white hindwing patch (A locus) and a region of approximately 5 Mb on Chromosome 15 (hereafter chr15) is associated with

both background orange/brown (B locus) and the forewing black tip (C locus) (Fig 2A and S2 Fig). Below, we refer to this region on chr15, which spans over 200 protein-coding genes, as the BC supergene [19], although we note that additional associated SNPs on Chromosome 22 suggest that background wing melanism may also be influenced by other loci.

Clustering analysis based on genetic distances reveals three clearly distinct alleles at the BC supergene (Fig 2C). This further supports the hypothesis of recombination suppression, although a number of individuals show mosaic ancestry consistent with occasional recombination (S4 Fig). The three main alleles correspond to the three common forewing phenotypes, so we term these $BC^{chrysippus}$ (orange background with black forewing tip, formerly *bbcc*), $BC^{dorippus}$ (orange without black tip, formerly *bbCC*), and $BC^{orientis}$ (brown background with black forewing tip, formerly *BBcc*) (Fig 2C). Fifteen of the twenty contact zone individuals are heterozygous, carrying two distinct BC alleles, and a few carry putative recombinant alleles, as do some of the southern African form orientis individuals (S4 Fig). As shown previously, $BC^{dorippus}$ (which includes the dominant C allele) and $BC^{orientis}$ (which includes the dominant B allele) are both dominant over the recessive $BC^{chrysippus}$ (Fig 2C and S4 Fig).

Although it can be challenging to identify particular functional mutations in regions of suppressed recombination, the presence of some recombinant individuals allowed us to narrow down candidate regions for the B and C loci. A cluster of SNPs most strongly associated with background colour (B locus) is found just upstream of the gene *yellow*, and a phylogenetic network for a 30-kb region around *yellow* groups individuals nearly perfectly by phenotype, although some individuals classed as heterozygous were intermingled with homozygotes (S5 Fig). In *Drosophila*, Yellow expression is associated with variation in melanism [37], and in some butterflies, *yellow* knockouts show reduced melanin pigmentation [38], making this a compelling candidate for the B locus. The strongest associations with forewing tip (C locus) occur at the gene *arrow*, and a phylogenetic network for a 100-kb region around this gene similarly clusters individuals by phenotype (S5 Fig). In *Drosophila*, Arrow is essential for Wnt signalling in wing development [39]. Wnt signalling is known to underlie variation in colour pattern in *Heliconius* butterflies [40], and knockout mutants for the Wnt ligand gene *WntA* in *D. plexippus* show a loss of pigmentation [41]. This makes *arrow* a promising candidate for the C locus. While these genes represent our best candidates, numerous strongly associated SNPs occurred closer to other genes in this region (S10 Table). Future studies will aim to narrow down and validate these associations.

Irrespective of their precise mode of action, the patterns of association imply that the B and C loci are approximately 1.6 Mb apart (S5 Fig) and would therefore be fairly loosely linked under normal recombination. This physical distance translates to around 7.6 cM, assuming crossover rates similar to those in *Heliconius* [31,42], whereas the estimated recombination distance between B and C based on crosses is 1.9 cM [43]. Theory predicts that recombination suppression can be favoured if it maintains linkage disequilibrium (LD) between co-adapted alleles in the face of gene flow [1–4]. Our study is one of only a few cases in which it can be shown that alleles at distinct loci that each influence a component of a complex trait are maintained in LD by suppressed recombination [44,45].

It is likely that chromosomal rearrangements contribute to recombination suppression at the BC supergene. Although our short-read data do not allow us to test directly for inversions, they do reveal dramatic variation in sequencing coverage over the proximal end of the chromosome. Comparison of coverage among individuals suggests a large (approximately 5 Mb) polymorphic insertion in this region that tends to occur in individuals carrying the $BC^{dorippus}$ allele (S6 Fig). Synteny comparison with *H. melpomene* reveals that this insertion involves an expansion in copy number of a region of several hundred kb. Comparison of copy numbers for two of the genes in the expansion with several other species confirms that it is derived in *D.*

*chrysippus* (S7 Fig). The expansion appears to occur just a few kb from the coding region of *arrow* (S6 Fig), and is also perfectly associated with the presence of the dominant dorippus phenotype (absence of black forewing tip) (S7 Fig). It is possible that it has a causal effect on the phenotype by influencing the expression of *arrow*, but it might simply be linked to the causative mutation. Either way, we suggest that this large structural change, which increases the length of the chromosome by nearly a third, contributes to recombination suppression between the $BC^{dorippus}$ allele and other supergene alleles by interfering with chromosome pairing in heterozygotes.

## A neo-W chromosome traps a single haplotype of chr15 in contact zone females

Previous crossing experiments indicated that the BC chromosome has become sex linked in contact zone females [22]. To confirm this hypothesis using genetic tools, we created a 'cured line' by treating a female from an all-female brood with tetracycline to eliminate *Spiroplasma* and allow the survival of male offspring [23]. A cross using this female confirms perfect sex-linkage of forewing phenotype ($n$ = 22, chi-squared test $p$ = 0.00002; S8 Fig). We then used PCR assays on a subsequent sibling cross from the cured line to confirm that maternal alleles for chr15 segregate with sex ($n$ = 22, $p$ < 0.00003), whereas paternal alleles segregate randomly ($n$ = 22, $p$ = 0.36; S8 Fig). These results exactly match the model (Fig 1B) in which the BC supergene has become linked to the W chromosome in females but continues to segregate as an autosome in males.

Although we were unable to definitively identify any scaffolds from the ancestral W chromosome, which is likely to be highly repetitive, we can test whether chr15 shows the expected hallmarks of a young neo-W, hypothesised to have formed through fusion to the ancestral W [22]. Due to the complete absence of recombination in females, we expect that a single fused haplotype of chr15 would be spreading in the population. Any unique mutations specific to this haplotype should therefore occur at high frequency in females and be absent in males. We scanned for such high-frequency female-specific mutations and found them to be abundant across the entire length of chr15 and nearly absent throughout the rest of the genome (Fig 3A). At the individual level, we can clearly identify 15 females (14 collected in the contact zone and the single 'cured line' female) that consistently share these high-frequency mutations (S9 Fig). Genetic distance among these females in the colinear region of chr15 (outside the BC supergene) is reduced, indicating that they all share a similar haplotype of the fused chromosome (Fig 3B).

## The neo-W formed recently and spread rapidly

Genetic variation accumulated in the neo-W lineage since its formation can tell us about its age. Sequence divergence between the neo-W and autosomal copies of chr15 (inferred from the density of heterozygous sites in the colinear region of chr15 in females carrying the neo-W) is not significantly different from that between the autosomal copies in 'wild-type' individuals that lack the fusion (Fig 3C, Wilcoxon signed rank test, $p$ = 0.36, $n$ = 48 windows of 100 kb each). This implies that insufficient time has passed since the fusion event for significant accumulation of new mutations. The limited divergence of the neo-W haplotype from the autosomal copy of chr15 in each female makes it challenging to isolate. Nonetheless, by using diagnostic mutations that are unique to and fixed in the neo-W lineage, we were able to identify sequencing reads from the shared haplotype and reconstruct a partial neo-W sequence for each female (S10 Fig). A dated genealogy based on these sequences places the root of the neo-

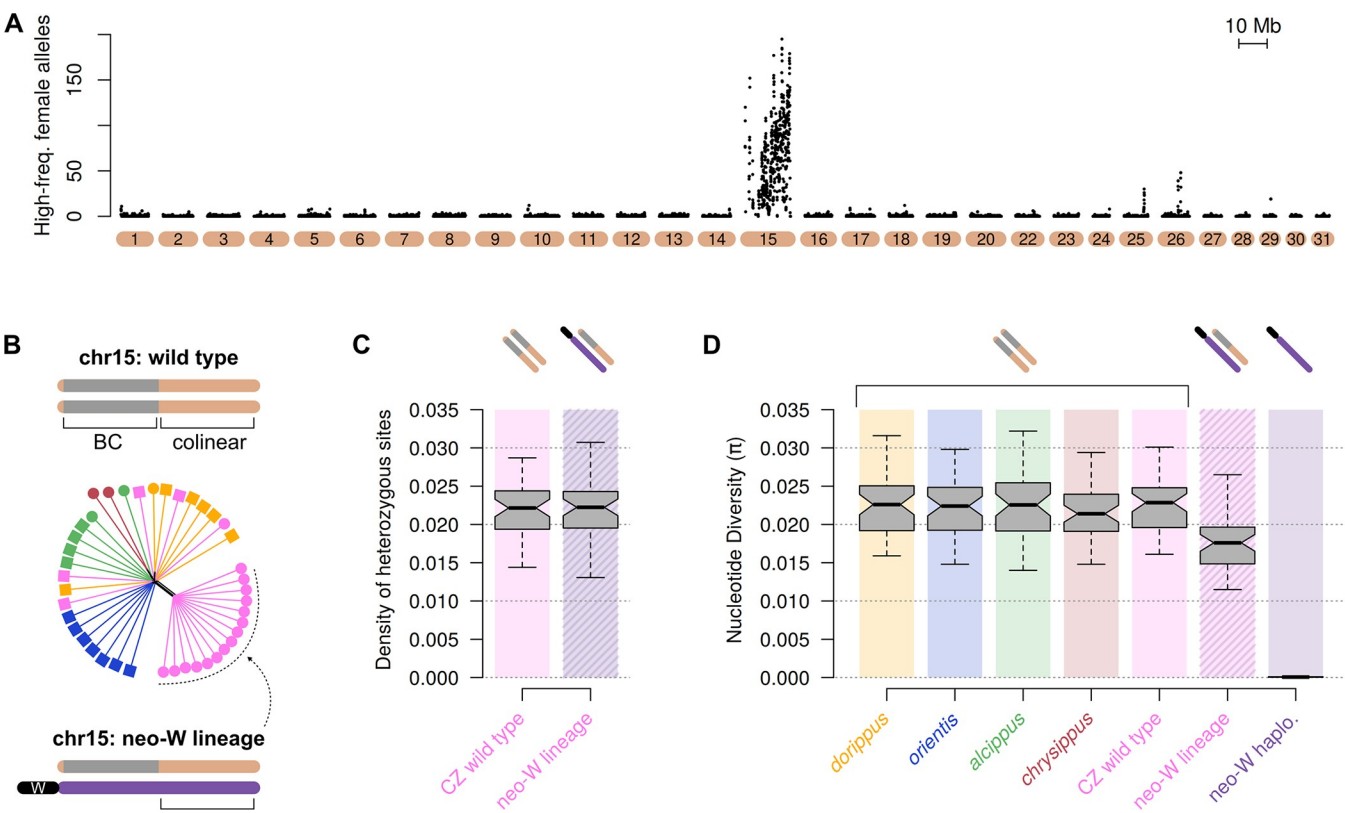

**Fig 3. Recent sweep of a young neo-W.** (**A**) The number of high-frequency female-specific mutations (>20% in females and absent in males) in 100-kb sliding windows (20-kb step size). (**B**) Distance-based phylogenetic network for the distal colinear region of chr15 outside of the BC supergene reveals that most contact zone females carry the conserved neo-W haplotype. Cartoons show how the colinear region of chr15 is outside of the BC supergene but would still capture reduced divergence among individuals carrying a shared non-recombining neo-W. (**C**) Box plot comparing the density of heterozygous sites in 100-kb windows in the colinear region of chr15 between wild-type individuals from the contact zone (CZ) and those carrying the neo-W. Cartoon chromosomes above the plot match those shown in panel B. A relative lack of elevated heterozygosity in the neo-W lineage indicates a lack of divergence of the fused neo-W haplotype, consistent with the fusion being recent. (**D**) Box plot of nucleotide diversity (π) within each population for the same colinear region of chr15. On the far right, π is shown for the haploid neo-W haplotype specifically, based on partial sequences isolated from this haplotype (see Methods and S10 Fig for details). The near absence of genetic diversity implies a very rapid spread of the neo-W through the population. Data deposited in the Dryad repository [36].

W lineage at approximately 2,200 years (26,400 generations) ago (posterior mean = 2,201, SD = 318).

The neo-W is present in all but one of the contact zone females, implying a rapid spread since its formation. This process is similar to a selective sweep of a beneficial mutation, except that complete recombination suppression in females means that the sweep affects the entire chromosome equally. Unlike a conventional sweep, it is not expected to eliminate genetic diversity from the population, as these females will also carry an autosomal copy of chr15 inherited from their father (Fig 1B). Indeed, we see a 20% reduction in overall nucleotide diversity (π) on chr15 in females of the neo-W lineage (Fig 3D). However, when we consider only the neo-W haplotype in each of these females, we see a nearly complete absence of genetic variation, with a π of 0.00007, more than two orders of magnitude lower than for autosomal copies of chr15 (0.0228) (Fig 3D). These results further support a very recent and rapid spread of the neo-W.

The neo-W haplotype carries the recessive $BC^{chrysippus}$ allele at the BC supergene (S4 Fig). However, previous work [22] shows that at the focal sampling site in the contact zone, most males are immigrants homozygous for the dominant $BC^{dorippus}$ allele, and the vast majority of

females (84%) are heterozygous $BC^{dorippus}/BC^{chrysippus}$, as expected if most inherit $BC^{dorippus}$ from their father and $BC^{chrysippus}$ (on the neo-W) from their mother. The dominant dorippus phenotype is therefore by far the most abundant in this population. Because aposematic colouration should be under positive frequency dependent selection, it is highly unlikely that the spread of the neo-W can be explained by selection on colour pattern, highlighting the question of what else might have driven its spread.

## Hitchhiking between the neo-W and *Spiroplama*

We hypothesised that the neo-W has spread as a result of co-inheritance with the male-killing *Spiroplasma*, which is itself spreading through the population as a selfish element. Experiments have suggested that all-female broods have enhanced survival relative to females from broods that include males, possibly due to reduced competition for resources [46], although other factors such as improved immunity [47] have not been tested. A similar boost to the relative fitness of infected females is thought to have driven the rapid spread of a male-killing *Wolbachia* in the butterfly *Hypolimnas bolina*, which has occurred over a similar timescale to that reported here [48]. For *Spiroplasma* to drive the spread of the neo-W, it would also need to be strictly vertically inherited down the female line, such that it is always co-inherited with the neo-W.

We identified nine scaffolds making up the 1.75-Mb *Spiroplasma* genome in our *D. chrysippus* assembly (S11 Fig). Infected individuals are clearly identifiable by mapping resequencing reads to the *Spiroplasma* scaffolds (S11 Fig), and this was confirmed by PCR. As predicted, all females in the neo-W lineage are infected (with the exception of the cured line female, in which *Spiroplasma* had been eliminated). Moreover, all infected females fall into the same mitochondrial clade (Fig 4A), consistent with matrilineal inheritance. To confirm that the *Spiroplasma* is strictly vertically inherited and always associated with a single female lineage, we used PCR assays for *Spiroplasma* and mitochondrial haplotype and expanded our sample size to 158 individuals, including samples used in previous studies going back two decades [19,23] (S12 Table and S12 Fig). This confirms the perfect association: 100% of infected individuals ($n = 42$) carry the same mitochondrial haplotype, and this haplotype is otherwise rare, occurring in 8% of uninfected individuals ($n = 116$) (S12 Fig).

Like the neo-W, the *Spiroplasma* genomes carry limited variation among individuals ($\pi = 0.0005$), consistent with a single and recent outbreak of the endosymbiont. Although the lack of variation makes it challenging to infer genealogies, our inferred maximum likelihood genealogies for the neo-W and *Spiroplasma* are strikingly congruent (Fig 4B). The low bootstrap support for multiple nodes is unsurprising, given that these sequences descend from a recent common ancestor, such that most nodes will be defined by only a few informative sites. This does not weaken the support for congruence, however, as the probability of two incorrectly inferred topologies matching by chance is infinitesimally small. In a permutation test for congruence between the two distance matrices [49], the observed level of congruence exceeds all 100,000 random permutations. There is therefore strong support for co-inheritance of the neo-W and *Spiroplasma* [50].

The combined spread of three physically unlinked DNA molecules—the mitochondrial genome, neo-W, and *Spiroplasma* genome—constitutes a form of genetic hitchhiking, but is facilitated by their strict matrilineal inheritance rather than physical linkage. We cannot entirely rule out the possibility that the neo-W is contributing to this spread, or even driving it entirely, through direct selection or meiotic drive. In theory, this is testable by examining broods that carry the neo-W but lack *Spiroplasma*, as these should comprise more females than males, despite the absence of the male-killer. We raised 11 such broods in our cured line,

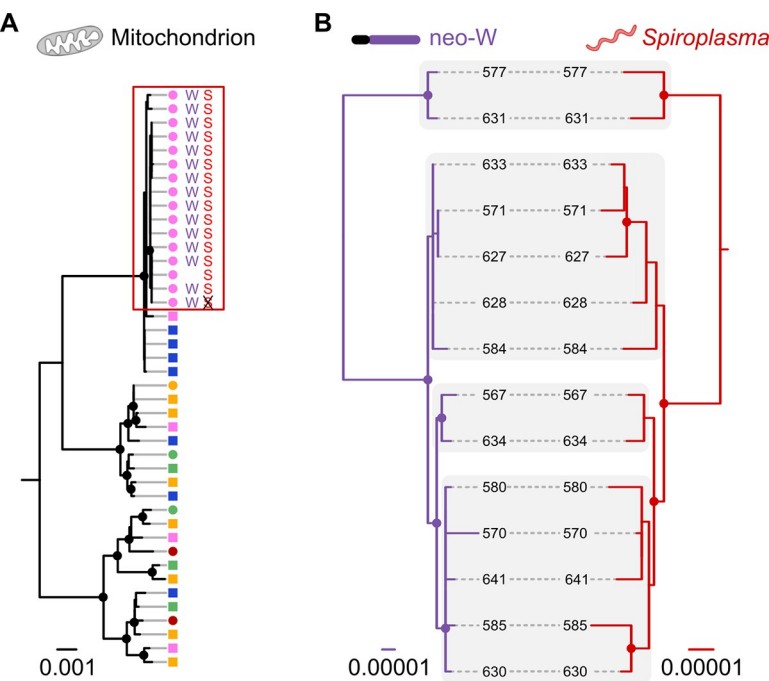

**Fig 4. Matrilineal inheritance causes coupling between neo-W and *Spiroplasma*.** (**A**) Maximum likelihood phylogeny for the whole mitochondrial genome. Individuals are coloured according to population of origin (see Fig 1A), and those carrying the neo-W ('W') and *Spiroplasma* ('S') are indicated (including one cured individual in which *Spiroplasma* was eliminated). Females are indicated by circles and males by squares. (**B**) Maximum likelihood phylogenies for the neo-W haplotype and *Spiroplasma* genome isolated from infected females. Corresponding clades are shaded to indicate congruence. Note that two samples are excluded in panel B: the cured sample, which lacked *Spiroplasma* because of tetracycline treatment, and one infected female found to lack the neo-W. Whether the latter represents an ancestral state or secondary loss requires further investigation. In all trees, nodes supported by more than 70 of 100 bootstrap replicates are indicated by circles. Data deposited in the Dryad repository [36].

and Smith [51] reported 10 natural broods that showed sex-linked colour pattern and no male-killing. Across these 21 broods, totalling 528 adult offspring, 51% were female. This is far from significantly different from the null expectation of 50% (binomial test $p = 0.7$). However, we note that to detect meiotic drive causing a 1% female bias with good power would require a far larger sample size of >15,000. Importantly, the few natural broods that have been found to show sex-linked colour pattern without male-killing have only been reported from regions in which *Spiroplasma* infection is present, implying that these broods result from occasional failed transmission of the endosymbiont [23]. Despite this potential for the neo-W to become decoupled from the male-killer, it has not spread beyond these regions, further supporting the hypothesis that hitchhiking with the male-killer underlies its rapid spread. Selfish elements have been shown to drive hitchhiking of the mitochondrial genome or a portion of a chromosome through a population and even across species boundaries [52–54]. Our findings show how an entire chromosome can be captured in the same way. Hitchhiking may therefore be of general importance in driving the spread of neo-sex chromosomes.

In *D. chrysippus*, it is currently unclear whether the neo-W or male-killer emerged first. It is also unclear whether their co-occurrence in a single ancestor was simply a coincidence or instead reflects some functional connection, such as the suggestion that the neo-W might confer susceptibility to the male-killer [22]. It is important to note that this is not the first time a neo-sex chromosome has formed in this lineage. A fusion of Chromosome 21 to the ancestral Z chromosome occurred in an ancestor of all *Danaus* species, producing a neo-Z [9,32,55]. It

is speculated that a complementary fusion of Chromosome 21 to the ancestral W also occurred [9,55], but this is difficult to conclusively verify because of degradation of the W chromosome over longer timescales. If this hypothesis of an ancient neo-W is correct, then the neo-W we describe (W-chr15) might in fact be better described as a neo-neo-W (W-chr21-chr15). It is possible that the spread of the original W-chr21 was also driven by hitchhiking with a selfish endosymbiont.

## Genetic and phenotypic consequences of recombination suppression

Sex chromosome evolution in many other taxa involves the progressive spread of recombination suppression outward from the sex-determining locus [56]. By contrast, the absence of crossing over in female meiosis means that a lepidopteran neo-W experiences complete and immediate recombination suppression over its entire length. Butterfly W chromosomes are therefore thought to be highly degenerated and repetitive, and to our knowledge none have been successfully assembled to date. The young age of the *D. chrysippus* neo-W therefore provides a rare opportunity to study the early evolutionary consequences of recombination suppression across an entire chromosome. Two related processes could shape its evolution: hitchhiking of preexisting deleterious mutations that were initially rare in the population [6], and accumulation of novel deleterious mutations due to reduced purging through recombination and selection (i.e., Muller's Ratchet) [7].

As a proxy for the 'genetic load' of deleterious mutations in the population, we considered $P_n/P_s$, the normalised ratio of non-synonymous to synonymous polymorphisms. Because of purifying selection, non-synonymous polymorphisms are typically rare, and where they do occur, the mutant allele typically occurs at low frequency in the population [57]. When considering all polymorphisms in the neo-W lineage, $P_n/P_s$ for chr15 (excluding the BC supergene, to avoid bias) is very slightly (approximately 5%) higher than for other autosomes (S13 Fig). Of 1,000 bootstrap replicates, 916 reproduced this bias, corresponding to a *p*-value of 0.084. However, when we partition polymorphisms by allele frequency, we see that chr15 carries a large excess of non-synonymous polymorphisms in the highest frequency class (i.e., minor allele at 50%), with a $P_n/P_s$ ratio >3 times larger than on other autosomes (S13 Fig). This holds across all 1,000 bootstrap replicates (i.e., $p < 0.001$). A change in the frequency distribution of non-synonymous variants, without a significant change in their abundance, is best explained by hitchhiking of preexisting mildly deleterious alleles that were initially rare in the population but were inadvertently carried to high frequency along with the neo-W haplotype, and are therefore now found in all females in this lineage. In fact, $P_n/P_s$ for high-frequency polymorphisms on chr15 is somewhat higher than would be expected through hitchhiking alone based on comparison with singleton mutations on other autosomes ($p = 0.044$). This suggests that accumulation of additional mildly deleterious alleles on the neo-W might have occurred early during its spread through the population.

At the phenotypic level, perhaps counterintuitively, the spread of a single supergene allele on the neo-W has not caused homogenisation of warning pattern among contact zone females and might in fact have the opposite effect. In locations where the neo-W and *Spiroplasma* are nearly fixed, such as our sampling site near Nairobi, the high incidence of male-killing implies that the population is strongly shaped by immigrant males. Because the $BC^{chrysippus}$ allele on the neo-W is universally recessive, daughters will tend to match the phenotype of their immigrant father. However, because the neo-W is always transmitted to daughters, the paternal chr15 copy will be lost to male-killing after one generation, creating a genetic sink for immigrant male genes [22] (S14 Fig). This combination of processes results in a female population that is highly sensitive to the source of immigrants, which is known to fluctuate seasonally

with monsoon winds [16,58] (S14 Fig). This model leads to the testable prediction that seasonal fluctuations in female phenotypes should be most dramatic where male-killing is most abundant.

### Future evolutionary trajectories

The future of the neo-W and *Spiroplasma* outbreak is uncertain. A lack of males could lead to local extinctions [27], but extinction of the entire infected lineage is unlikely given the high dispersal ability and seasonal influxes of males in the contact zone. Indeed, it is notable that *Spiroplasma* infection has only been recorded within the contact zone population (with the exception of a single South African brood reported here, S12 Table), especially given theory showing that male-killers should spread very rapidly across the geographical range of a panmictic population if they provide even a very weak selective advantage [48]. Future work will investigate whether its spread might be curtailed by environmental factors, for example if oviposition behaviour or host plant availability only leads to sibling competition (and consequent benefits for all-female broods) under certain conditions [46]. An alternative and non–mutually exclusive hypothesis is that dispersal rates of infected females are strongly reduced. In other systems, sex-ratio distortion has driven adaptive responses by the host, including changes to the mating system [59] and the evolution of resistance to male-killing [60,61]. The absence of evidence for these phenomena in *D. chrysippus* might simply reflect the recency of the male-killing outbreak. Eventually, we also expect the non-recombining neo-W to begin to degenerate through further hitchhiking, gene loss, and the spread of repetitive elements [8,56]. This young system provides a rare opportunity to study how these phenomena unfold through time and space.

## Methods

### Ethics statement

Butterfly collection was performed under permit where relevant: NACOSTI/P15/3290/3607, NACOSTI/P15/2403/3602 (National Commission for Science and Technology, Kenya), MINEDUC/S&T/459/2017 (Ministry of Education, Rwanda), EMDEP006/17 (Environmental Management Division, St Helena Government); and always with permission of the land owner and/or local authorities. We also worked with local researchers wherever possible, including authors DJM, KSO, SC, and IJG and with the Lepidopterists Society of Africa.

### Reference genome sequencing, assembly, and annotation

Detailed methods for generation of the *D. chrysippus* reference genome are provided in S1 Text. Briefly, a draft assembly was generated using SPAdes [62] from a combination of paired-end and mate-pair libraries of various insert sizes. Scaffolding and resolution of haplotypes was performed using Redundans [63] and Haplomerger2 [64]. The assembly was annotated using a combination of de novo gene predictors, yielding 16,654 protein coding genes. Mitochondrial genomes were assembled using NOVOplasty [65].

Although we currently lack linkage information for further scaffolding, we generated a pseudo-chromosomal assembly based on homology with the highly contiguous *H. melpomene* genome [30,31,66], adjusted for known karyotypic differences [9,30–32,55]. Although these genomes are diverged by approximately 90 million years, this homology-based approach has been shown previously to be successful for reconstructing chromosomes in a fragmented *D. plexippus* genome [9]. In total, 282 Mb (87% of the genome) could be confidently assigned to chromosomes (S1 Fig).

Scaffolds representing the *Spiroplasma* genome were identified based on read depth of remapped reads (S11 Fig) and homology to other available *Spirolasma* genomes. Annotation was performed using the RAST server pipeline [67,68].

### Population sample resequencing and genotyping

This study made use of 42 newly sequenced *D. chrysippus* individuals, as well as previously sequenced individuals of the sister species, *D. petilia* ($n$ = 1) and the next closest outgroup, *D. gilippus* ($n$ = 2) [69] (S9 Table). Details of DNA extraction, sequencing, and genotyping are provided in S1 Text. Briefly, DNA was extracted from thorax tissue and sequenced (paired-end, 150 bp) to a mean depth of coverage 20× or greater. Reads were mapped to the *D. chrysippus* reference assembly using Stampy [70] v1.0.31, and genotyping was performed using GATK version 3 [71,72]. Genotype calls were required to have an individual read depth $\geq$8, and heterozygous and alternate allele calls were further required to have an individual genotype quality (GQ) $\geq$20 for downstream analyses.

### Genomic differentiation and associations with wing pattern

We used the fixation index ($F_{ST}$) and absolute divergence ($d_{XY}$) to examine genetic differentiation across the genome among the three subspecies for which we had six or more individuals sequenced. $F_{ST}$ and $d_{XY}$ were computed using the script popgenWindows.py (github.com/simonhmartin/genomics_general release 0.2) with a sliding window of 100 kb, stepping in increments of 20 kb. Windows with fewer than 20,000 genotyped sites after filtering (see above) were ignored.

To identify SNPs associated with the three Mendelian colour pattern traits (i.e., the A, B, and C loci) (Fig 1A), we used PLINK v1.9 [73] with the '—assoc' option and provided quantitative phenotypes of 0, 1, or 0.5 for assumed heterozygotes, which causes PLINK to use the Wald test for quantitative traits. In addition to the quality and depth filters above, SNPs used for this analysis were required to have genotypes for at least 40 individuals, a minor allele count of at least 2, and to be heterozygous in no more than 75% of individuals. SNPs were also thinned to a minimum distance of 100 bp.

To examine relationships among diploid individuals in specific regions of interest, we constructed phylogenetic networks using the Neighbor-Net [74] algorithm, implemented in SplitsTree [75]. Pairwise distances used for input were computed using the script distMat.py (github.com/simonhmartin/genomics_general release 0.2).

### Haplotype cluster assignment

To assign haplotypes to clusters in the BC supergene region, we first phased genotypes using SHAPEIT2 [76,77] using SNPs filtered as for association analysis above, except with a minor allele count of at least 4 and no thinning. Default parameters were used for phasing except that the effective population size was set to $3 \times 10^6$. To minimise phasing switch errors, we analysed each 20-kb window separately. Cluster assignment for both haplotypes from each individual was based on average genetic distance to all haplotypes from each of three reference groups: *D. c. dorippus*, *D. c. orientis*, or *D. c. alcippus* (the latter is also representative of *D. c. chrysippus*, as they share the same alleles at the BC supergene). A haplotype was assigned to one of the three groups if its average genetic distance to members of that group was less than 80% of the average distance to the other two groups; otherwise, it was left as unassigned. Genetic distances were computed using the script popgenWindows.py (github.com/simonhmartin/genomics_general release 0.2).

## Identification of neo-W–specific sequencing reads

To identify females carrying the neo-W chromosome, we visualised the distribution of female-specific derived mutations that occur at high frequency. Allele frequencies were computed using the script freq.py (github.com/simonhmartin/genomics_general release 0.2). Because of the absence of female meiotic crossing over in Lepidoptera, all females carrying the neo-W fusion should share a conserved chromosomal haplotype for the entire fused chromosome. To isolate this shared fused haplotype from the autosomal copy, we first identified diagnostic mutations as those that are present in a single copy in each member of the 'neo-W lineage' and absent from all other individuals and outgroups. We then isolated the sequencing read pairs from each of these females that carry the derived mutation (S10 Fig). This resulted in a patchy alignment file, with a stack of read pairs over each diagnostic mutation. Based on these aligned reads, we genotyped each individual as described above, except here setting the ploidy level to 1, and requiring a minimum read depth of 3.

## Diversity and divergence of the neo-W

The lack of recombination across the neo-W makes it possible to gain insights into its age. Over time, mutations will arise that differentiate the neo-W from the recombining autosomal copies of the chromosome. We estimated this divergence based on average heterozygosity in females carrying the neo-W and compared it to heterozygosity from contact-zone individuals not carrying the neo-W. Heterozygosity was computed using the script popgenWindows.py (github.com/simonhmartin/genomics_general release 0.2), focusing only on the colinear portion of the chromosome (i.e., the distal portion from 11 Mb onwards), which is outside of the BC supergene. Heterozygosity was computed in 100-kb windows, and windows were discarded if they contained fewer than 20,000 sites genotyped in at least two individuals from each population.

A recent spread of the neo-W through the population should also be detectable in the form of strong conservation of the neo-W haplotype in all females that carry it (i.e., reduced genetic diversity). We therefore computed nucleotide diversity ($\pi$) in 100-kb windows as above. Reported values of $\pi$ and heterozygosity represent the mean ± standard deviation across 100-kb windows.

## Genealogical analyses

We produced maximum likelihood trees for the mitochondrial genome, neo-W, and *Spiroplasma* genome, using PhyML v3 [78] with the GTR substitution model. Given the small number of SNPs in both the neo-W and *Spiroplasma* genomes, regions with inconsistent coverage across individuals were excluded manually. Only sites with no missing genotypes were included.

We estimated the root node age for the neo-W using BEAST2 [79,80] version 2.5.1 with a fixed clock model and an exponential population growth prior. For all other priors we used the defaults as defined by BEAUti v2.5.1. We assumed a mutation rate of $2.9 \times 10^{-9}$ per generation based on a direct estimate for *Heliconius* butterflies [81] and 12 generations per year [82]. BEAST2 was run for 500,000,000 iterations, sampling every 50,000 generations, and we used Tracer [83] version 1.7.1 to check for convergence of posterior distributions and compute the root age after discarding a burn-in of 10%.

We tested for congruence between the neo-W and *Spiroplasma* trees using PACo [49], which assesses the goodness of fit between host and parasite distance matrices, with 100,000 permutations. Distance matrices were computed using the script distMat.py (github.com/simonhmartin/genomics_general release 0.2).

## Analysis of synonymous and non-synonymous polymorphism

We computed $P_n/P_s$ as as the ratio of non-synonymous polymorphisms per non-synonymous site to synonymous polymorphisms per synonymous site. Synonymous and non-synonymous sites were defined conservatively as 4-fold degenerate and 0-fold degenerate codon positions, respectively, with the requirement that the other two codon positions are invariant across the entire dataset. Only sites genotyped in all 15 females in the neo-W lineage were considered, and counts were stratified by minor allele frequency using the script sfs.py (github.com/simonhmartin/genomics_general release 0.1).

## Butterfly rearing and molecular diagnostics

To generate a stock line that is cured of *Spiroplasma* infection, we treated caterpillars from all-female broods with tetracycline, following Jiggins and colleagues [23]. A 'cured line' was initiated from a single treated female that had the heterozygous *Cc* transiens phenotype (Fig 1A). This female was crossed to a wild male (homozygous *cc*) to test for sex linkage of phenotype. The cured line was maintained through sibling crosses for six generations and the persistence of males indicated that *Spiroplasma* had been eliminated.

We then applied a molecular test for sex linkage of chr15 using the F5 brood from the cured line. We designed two separate PCR diagnostics based on SNPs segregating on chr15 to distinguish between the two chromosomes of the male and the female parents (S11 Table). PCR was performed using the Phusion HF Master Mix and HF Buffer (New England Biolabs, Ipswich, MA).

To screen for *Spiroplasma* infection, we designed a PCR assay targeting the glycerophosphoryl diester phosphodiesterase (GDP) gene (S11 Table). PCR was performed as above. We confirmed the sensitivity of this diagnostic by analysing individuals of known infection status based on whole genome sequencing (12 infected and 11 uninfected).

To investigate whether *Spiroplasma* infection was always associated with a single mitochondrial haplotype, we designed a PCR RFLP for the Cytochrome Oxidase Subunit I (COI) that differentiates the infected 'K' lineage (S11 Table). PCR was performed as above. A subset of products were verified by Sanger sequencing after purification using the QIAquick PCR Purification Kit (Qiagen).

## Supporting information

**S1 Fig. Pseudo-chromosomal assembly of *D. chrysippus*.** Homology with the *H. melpomene* genome (corrected for known fusion events [9,30,32] and scaffolded into chromosomes [31]) (blue) allowed us to construct a robust pseudo-chromosomal assembly for *D. chrysippus*. Scaffolds of *D. chrysippus* are shown in alternating shades of orange. Blue lines connect homologous genes (BLAST E-value < $1 \times 10^{-20}$, identity >50%). Data deposited in the Dryad repository [36].
(PNG)

**S2 Fig. Genetic differentiation and SNP associations with colour pattern.** $F_{ST}$ is plotted across each chromosome between three different subspecies of *D. chrysippus*, as indicated above the plot. Scaffolds are indicated by light and dark shading. Numbers on the x-axis indicate chromosome position in Mb. Coloured crosses above the plots indicate SNPs strongly associated with the phenotypes controlled by the A (grey), B (brown), and C (black) loci (Wald test, 99.99% quantile). A number of candidate genes are annotated on the plot. These include known and putative wing patterning genes in *Heliconius* (*optix* [84], *cortex* [85], *WntA* [40], *aristaless* [86], and *ventral veins lacking* [42]) and *Papilio* spp. (*doublesex* [87] and *engrailed*

[88]). A *myosin* gene thought to be associated with a pale mutant form in *D. plexippus* [69] is also indicated, along with *collegen type IV*, which was found to be associated with migratory behaviour in *D. plexippus* [69]. Several melanism-related genes are also annotated, as well as *arrow*, which was added to the list of candidates post hoc due to strong association with colour pattern (see main text). Of our a priori candidates, only *yellow* is found to associate with colour pattern in *D. chrysippus*. Data deposited in the Dryad repository [36].
(PNG)

**S3 Fig. $d_{XY}$ plotted against π reveals excess divergence on colour pattern–associated chromosomes.** Absolute divergence between each pair of populations ($d_{XY}$) and nucleotide diversity within populations (π) were computed for nonoverlapping 100-kb windows. The value of π plotted is the average between the two populations in each plot. The clustering of points along the diagonal indicates that diversity within each subspecies is similar to divergence between subspecies, consistent with a single nearly panmictic population. Points that deviate to the left of the diagonal indicate either excess divergence between subspecies or reduced diversity within subspecies, or both. Here, the colour pattern–associated regions on Chromosomes 4 and 15 (indicated in colour for convenience) show signatures of local adaptation with both reduced within-population diversity and increased between-population divergence, as would be expected if selection limits effective gene flow at these loci. One pair of populations, *D. c. dorippus* and *D. c. orientis*, are diverged at chr15 but not Chromosome 4, which is also expected as they only differ in their forewing phenotype and both lack the white hindwing patch. Data deposited in the Dryad repository [36]. chr15, Chromosome 15.
(PNG)

**S4 Fig. Allelic clustering across chr15 for all samples.** Coloured blocks indicate 20-kb windows in which sequence haplotypes could be clustered into one of three genetic clusters (yellow: dorippus, red: chrysippus/alcippus, blue: orientis) based on pairwise genetic distances (see Methods for details). Windows in grey show insufficient relative divergence to be assigned to a cluster. White gaps indicate missing data. There are three clearly distinct alleles that correspond largely with colour pattern. Heterozygotes indicate a dominance hierarchy: The $BC^{dorippus}$ allele (yellow) is the most dominant and produces the dorippus phenotype (no black forewing tip). Around half of the heterozygotes with one copy of the dorippus allele express the transiens phenotype, with white marks on the forewing. The $BC^{orientis}$ allele (blue) corresponds with the orientis phenotype (black wing tip and dark background colour). It is dominant over the $BC^{chrysippus}$ allele, which produces the chysippus phenotype (black wing tip with light background colour) only when homozygous. There is evidence of recombination in the form of mosaic haplotypes. Finally, samples found to be carrying the neo-W chromosome (see main text) are indicated with an asterisk. All carry the $BC^{chrysippus}$ allele. Note that no phenotype was recorded for the reference genome individual RF.K001. Data deposited in the Dryad repository [36]. chr15, Chromosome 15.
(PNG)

**S5 Fig. Candidate loci for forewing colour pattern on chr15.** Differentiation ($F_{ST}$) is plotted across part of chr15 (bottom). Above the plot, locations of SNPs most strongly associated with the B and C loci (Wald test, 99.99% quantile) are shown: 'B locus' (controlling brown/orange background) in brown and 'C locus' (controlling forewing black tip) in black. The best respective candidate genes, *yellow* and *arrow*, are indicated on the plot. At the top, distance-based phylogenetic networks constructed for regions around the candidate genes (30 kb around *yellow* and 100 kb around *arrow*) are shown. Colours indicate subspecies as in Fig 1A, and shapes indicate sex. Phenotypes are coded black and white for putative homozygotes and grey for

putative heterozygotes. A corresponding network for the whole genome is included for comparison, showing how undifferentiated the subspecies are in general. Data deposited in the Dryad repository [36]. chr15, Chromosome 15.
(PNG)

**S6 Fig. Variable coverage reveals a large expansion on chr15.** (**A**) Dots indicate median read coverage in 20-kb windows across chr15, normalised relative to the genome-wide mean (dashed line). Twelve representative individuals are shown. All individuals fall into one of three categories: normal coverage, approximately half coverage, or approximately zero coverage across the first third (5.84 Mb) of the chromosome, indicating an insertion polymorphism that is either homozygous present/absent or heterozygous. Coloured blocks indicate allelic clustering for each 20-kb window (see S4 Fig), with white indicating gaps in the alignment because of variable sequence coverage. (**B**) Comparison of homologous genes in the *H. melpomene* genome indicates several genes near the proximal end of the chromosome that are duplicated multiple times in our *D. chrysippus* reference genome. Locations of the candidate B and C genes *yellow* and *arrow* (see S5 Fig) are indicated. Scaffolds in the *D. chrysippus* pseudo-chromosomal assembly are alternately shaded light and dark. Data deposited in the Dryad repository [36]. chr15, Chromosome 15.
(PNG)

**S7 Fig. An expansion in the *BC^dorippus* allele of chr15 involves multiple gene duplications.**
(**A**) Depth of coverage across the expansion region (see S6 Fig), in each individual, normalised by the genome average. Points represent the median coverage over 20-kb windows, and vertical lines indicate the 25% and 75% quantiles. Homozygous individuals with two copies of the expansion have a normal depth of approximately 1, heterozygous individuals have a depth of approximately 0.5, and those homozygous for a lack of the expansion have a depth of approximately 0. There is perfect correspondence between presence of the expansion and the dorippus phenotype (lack of black forewing tip). Heterozygotes display either the dorippus pattern or the transiens pattern, with white marks on the forewing, consistent with the approximately 50% penetrance described in previous crosses [89]. (**B**) Maximum likelihood phylogeny of Nephrin-like protein sequences encoded by two genes located within the expansion region. Homologous genes from *Danaus plexippus*, *H. melpomene*, and *Melitaea cinxia* are included. The tree indicates that the ancestral state in the Nymphalidae is to have two copies of the gene, while the *D. chrysippus* assembly has 14 copies (8 and 6, respectively). (**C**) The number of copies of *nephrin-like 1* and *2* is indicated in black and grey, respectively. Although we have just one assembly from a sample homozygous for the *BC^dorippus* allele, the read-depth data (see panel A and S6 Fig) suggest that the other *D. chrysippus* morphs have the ancestral state, lacking the additional copies, as do the two outgroup species: *D. petilia* and *D. gilippus*. Data deposited in the Dryad repository [36]. chr15, Chromosome 15.
(PNG)

**S8 Fig. Sex-linked inheritance of colour pattern and chr15 in a cured line.** (**A**) Sex linkage of forewing pattern controlled by the BC supergene. A female descending from the contact zone (top left) was cured of *Spiroplasma*. Her transiens phenotype indicated that she was heterozygous *Cc* (Fig 1B). She was crossed with a *cc* male (black forewing tips) to produce the F1 brood shown. Male offspring (right) who would ordinarily have been killed by *Spiroplasma* expressed the dorippus (or transiens) phenotype without black forewing tips, indicating that they had all inherited the *C* allele from their mother (note that males can be identified by the additional large black spot on the hindwing). Female offspring (left) all expressed the chrysippus phenotype, indicating that they had inherited the recessive *c* allele from both parents. (**B**)

Inheritance of two chr15 PCR markers (here designated P and Q) was tracked in the F5 brood of the cured line. One marker ('P') was heterozygous in the mother and showed complete sex linkage. The other marker ('Q') was heterozygous in the father and segregated independently of sex. These results are consistent with chr15 forming a neo-W in the mother, while both copies of the father's chr15 are autosomal. chr15, Chromosome 15.
(PNG)

**S9 Fig. Distribution of female-specific mutations identifies the neo-W lineage.** The 30 chromosomes are shown with each line representing an individual, coloured according to population: yellow = *D. c. dorippus*, red = *D. c. chrysippus*, green = *D. c. alcippus*, blue = *D. c. orientis*, pink = contact zone. Black points indicate the location of mutations shared by at least four females and absent from males. These are strongly clustered on chr15 and shared by a group of contact zone females, indicating that a conserved neo-W haplotype is shared by this female lineage. The noticeable absence of mutations on the proximal (left) region of chr15 reflects the large sequencing gaps corresponding to the expansion cluster in the $BC^{dorippus}$ allele (see S6 Fig). Data deposited in the Dryad repository [36].
(PNG)

**S10 Fig. Identification of mutations and sequence reads specific to the neo-W.** Schematic representation of the bioinformatic pipeline to isolate the neo-W haplotype from unphased resequencing data. Due to the recency of its formation, sequencing reads from the neo-W are not significantly divergent and will therefore map to the reference genome chr15. The challenge is to separate reads that derive from the neo-W and autosomal haplotypes, despite them all mapping to the same parts of the reference genome. Our solution is to use diagnostic mutations that are unique to the neo-W haplotype and shared by the multiple individuals that carry the neo-W. We identified candidate mutations specific to the neo-W haplotype as those at which all 15 females in the neo-W lineage are heterozygous, while all 27 remaining individuals are homozygous. We then used these candidate neo-W specific mutations to extract sequence reads that are specific to the neo-W. These represent only a fraction of the chromosome, because they represent only the reads carrying diagnostic mutations and their paired-end partners. The identification of these neo-W specific reads allows the identification of additional mutations on the same read that occurred after the formation of the neo-W. These can be used to estimate genetic diversity across the neo-W (accounting for the large amount of missing data) and also to infer a genealogy for the neo-W.
(PNG)

**S11 Fig. Identification of *Spiroplasma* genome and infection status based on read depth.** (**A**) Sequencing read depth of coverage averaged by scaffold (y-axis, exponential scale) and plotted against scaffold length (x-axis, log scale). Depth is shown for a suspected infected female above and a female from the tetracycline-treated 'cured line' below. Scaffolds identified as belonging to the *Spiroplasma* genome are shown in red. The mitochondrial genome is shown in blue. (**B**) Bars show the average depth of reads mapping to the *Spiroplasma* genome for each resequenced *D. chrysippus* individual. Note that all females from the hybrid zone are found to be infected, with the exception of the single individual from the cured line. Data deposited in the Dryad repository [36].
(PNG)

**S12 Fig. Association between mitochondrial haplotype and *Spiroplasma* infection.** (**A**) A whole mitochondrial maximum-likelihood phylogeny for the 42 resequenced individuals indicates that all infected *D. chrysippus* females belong to a single mitochondrial clade (here called the K lineage), consistent with strict matrilineal inheritance of *Spiroplasma*. Note that the

single *D. petilia* male from Australia was found to be infected by a related *Spiroplasma* strain but has a different mitochondrial haplotype, indicating an independent infection. (**B**) COI haplotype network for 66 individuals further supports the finding that only K lineage individuals are infected. (**C**) A PCR assay (see S11 Table) for an SNP specific to the K lineage applied to 158 individuals further confirms the finding that only the K lineage carries the infection. Note that one male was found to be infected, probably representing a rare survivor from an infected mother, as has been observed in some experimental crosses [23]. Data deposited in the Dryad repository [36]. COI, Cytochrome Oxidase Subunit I.
(PNG)

**S13 Fig. Evidence for hitchhiking of non-synonymous mutations on the neo-W.** Barplots (top) show the frequency distribution of synonymous (grey) and non-synonymous (black) polymorphisms in the neo-W lineage (i.e., contact-zone females carrying the neo-W chromosome). Values for chr15 are shown on the left and combined values across all other autosomes are shown on the right. Below, $P_n/P_s$ (the normalised ratio of non-synonymous to synonymous polymorphisms) is shown for each frequency class. Error bars show the 95% confidence interval based on 1,000 bootstrap replicates. These plots show that non-synonymous polymorphisms are generally skewed toward lower frequency but that chr15 carries a significant excess of non-synonymous polymorphisms at high frequency in the population. This is consistent with hitchhiking of previously rare mildly deleterious alleles to high frequency on the neo-W. Data deposited in the Dryad repository [36]. chr15, Chromosome 15.
(PNG)

**S14 Fig. Seasonal migration and a genetic sink drive fluctuations in local wing pattern.** (**A**) Average monthly frequencies of the black forewing phenotype (*cc* genotype, $BC^{orientis}$ and $BC^{chrysippus}$ alleles) show how immigration of different subspecies into the contact zone varies seasonally (data from Smith and colleagues [16], collected at Dar es Salaam between 1972 and 1975). (**B**) Phenotypes of females carrying the neo-W and *Spiroplasma* depend on the source of immigrant males (top row). Each generation, females (middle row) inherit both the neo-W and *Spirplasma* from their mother, and an autosomal chr15 copy from their immigrant father. The neo-W is recessive, causing these females to express their father's phenotype. After persisting in the female for one generation, the autosomal chr15 copy carrying the paternal allele is lost through male-killing, i.e., a genetic sink (bottom row). The progression from left to right illustrates how seasonal changes in the predominant source of immigrant males can drive corresponding changes in the phenotypes of the contact zone females. Data deposited in the Dryad repository [36]. chr15, Chromosome 15.
(PNG)

**S1 Table. Sequence data used for reference genome assembly.**
(PDF)

**S2 Table. Inferred genome properties based on k-mer content.**
(PDF)

**S3 Table. Final *D. chrysippus* assembly statistics.**
(PDF)

**S4 Table. Summarized results of the CEGMA analysis based on 248 CEGs.** CEG, Core Eukaryotic Genes; CEGMA, Core Eukaryotic Genes Mapping Approach.
(PDF)

**S5 Table. BUSCO statistics for 3 clades.**
(PDF)

**S6 Table. Summary of gene features in *D. chrysippus* genome.**
(PDF)

**S7 Table. Orthogroups summary statistics.**
(PDF)

**S8 Table. Distribution of orthogroups in different species.**
(PDF)

**S9 Table. Sample information for population genomic analyses.**
(PDF)

**S10 Table. Closest genes to SNPs most strongly associated with colour pattern traits.** Shading indicates the best candidate gene(s) with the most nearby associated SNPs.
(PDF)

**S11 Table. Details of genotyping assays.**
(PDF)

**S12 Table. Mitochondrial haplotype and infection status of 158 samples screened.** Screening for mitochondrial type was either through direct sequencing or PCR RFLP for a diagnostic SNP in the COI amplicon. Screening for infection status was either based on resequencing data (see S11 Fig) or by PCR amplification of the *Spiroplasma* GDP gene. COI, Cytochrome Oxidase Subunit I; GDP, glycerophosphoryl diester phosphodiesterase; RFLP, restriction fragment length polymorphism.
(PDF)

**S1 Text.**
(PDF)

## Acknowledgments

We are grateful to Godfrey Amoni Etelej, Laura Hebberecht-Lopez, and Glennis Julian for support with butterfly rearing. We thank Roger Vila, Frank Jiggins, and David Pryce for providing samples, and Jenny York, Frank Jiggins, Deborah Charlesworth, and Greg Hurst for helpful comments.

## Author Contributions

**Conceptualization:** Simon H. Martin, Ian J. Gordon, Steve Collins, Walther Traut, David A. S. Smith, Chris D. Jiggins, Richard H. ffrench-Constant.

**Data curation:** Simon H. Martin, Kumar Saurabh Singh.

**Formal analysis:** Simon H. Martin, Kumar Saurabh Singh.

**Funding acquisition:** Simon H. Martin, Ian J. Gordon, David A. S. Smith, Chris D. Jiggins, Chris Bass, Richard H. ffrench-Constant.

**Investigation:** Simon H. Martin, Kumar Saurabh Singh, Kennedy Saitoti Omufwoko, Steve Collins, Ian A. Warren, Hannah Munby, Walther Traut.

**Methodology:** Simon H. Martin, Kumar Saurabh Singh, Kennedy Saitoti Omufwoko, Steve Collins, Ian A. Warren, David A. S. Smith.

**Project administration:** Simon H. Martin, Ian J. Gordon, Dino J. Martins, David A. S. Smith, Richard H. ffrench-Constant.

**Resources:** Ian J. Gordon, Oskar Brattström, Dino J. Martins, Chris D. Jiggins, Richard H. ffrench-Constant.

**Software:** Simon H. Martin.

**Supervision:** Simon H. Martin, Dino J. Martins, Chris D. Jiggins, Chris Bass.

**Validation:** Simon H. Martin.

**Visualization:** Simon H. Martin.

**Writing – original draft:** Simon H. Martin, Kumar Saurabh Singh, Ian J. Gordon, Richard H. ffrench-Constant.

**Writing – review & editing:** Simon H. Martin, Ian J. Gordon, Walther Traut, David A. S. Smith, Chris D. Jiggins, Chris Bass, Richard H. ffrench-Constant.

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
