## [Editor Report · Decision Letter 0]

8 Aug 2019

Dear Simon, 

Thank you for submitting your manuscript entitled "Whole-chromosome hitchhiking driven by a male-killing endosymbiont" for consideration as a Research Article by PLOS Biology.

Your manuscript has now been evaluated by the PLOS Biology editorial staff, as well as by an academic editor with relevant expertise, and I'm writing to let you know that we would like to send your submission out for external peer review.

*Please be aware that, due to the voluntary nature of our reviewers and academic editors, manuscripts may be subject to delays during the holiday season. Thank you for your patience.*

Please re-submit your manuscript within two working days, i.e. by Aug 12 2019 11:59PM.

Best wishes,

Roli

Senior Editor

PLOS Biology

---

## [Decision Letter · Decision Letter 1]

13 Sep 2019

Dear Simon,

Many thanks very much for submitting your manuscript "Whole-chromosome hitchhiking driven by a male-killing endosymbiont" for consideration as a Research Article at PLOS Biology. Your manuscript has been evaluated by the PLOS Biology editors, an Academic Editor with relevant expertise, and by three independent reviewers.

You'll see that the reviewers are broadly positive, but they all request some textual and presentational changes, plus a few analyses. In addition, reviewer #2 has some more substantial requests, which the Academic Editor asks that you address with experimental data where appropriate.

IMPORTANT: While you submitted this paper as a full Research Article, we feel that it would be better considered as a Short Report. This is largely a cosmetic/editorial issue, but because the format has a maximum number of 4 main Figures, you will need to reduce your number of Figs by one. You can do this either by combining two existing Figs or by moving one to the Supplement. Please also select the article type "Short Report" when you re-submit.

In light of the reviews (below), we will not be able to accept the current version of the manuscript, but we would welcome resubmission of a much-revised version that takes into account the reviewers' comments. We cannot make any decision about publication until we have seen the revised manuscript and your response to the reviewers' comments. Your revised manuscript is also likely to be sent for further evaluation by the reviewers.

Your revisions should address the specific points made by each reviewer. Please submit a file detailing your responses to the editorial requests and a point-by-point response to all of the reviewers' comments that indicates the changes you have made to the manuscript. In addition to a clean copy of the manuscript, please upload a 'track-changes' version of your manuscript that specifies the edits made. This should be uploaded as a "Related" file type. You should also cite any additional relevant literature that has been published since the original submission and mention any additional citations in your response. 

Before you revise your manuscript, please review the following PLOS policy and formatting requirements checklist PDF: http://journals.plos.org/plosbiology/s/file?id=9411/plos-biology-formatting-checklist.pdf. It is helpful if you format your revision according to our requirements - should your paper subsequently be accepted, this will save time at the acceptance stage.

Please note that as a condition of publication PLOS' data policy (http://journals.plos.org/plosbiology/s/data-availability) requires that you make available all data used to draw the conclusions arrived at in your manuscript. If you have not already done so, you must include any data used in your manuscript either in appropriate repositories, within the body of the manuscript, or as supporting information (N.B. this includes any numerical values that were used to generate graphs, histograms etc.). For an example see here: http://www.plosbiology.org/article/info%3Adoi%2F10.1371%2Fjournal.pbio.1001908#s5.

For manuscripts submitted on or after 1st July 2019, we require the original, uncropped and minimally adjusted images supporting all blot and gel results reported in an article's figures or Supporting Information files. We will require these files before a manuscript can be accepted so please prepare them now, if you have not already uploaded them. Please carefully read our guidelines for how to prepare and upload this data: https://journals.plos.org/plosbiology/s/figures#loc-blot-and-gel-reporting-requirements.

Upon resubmission, the editors will assess your revision and if the editors and Academic Editor feel that the revised manuscript remains appropriate for the journal, we will send the manuscript for re-review. We aim to consult the same Academic Editor and reviewers for revised manuscripts but may consult others if needed.

We expect to receive your revised manuscript within two months. Please email us (plosbiology@plos.org) to discuss this if you have any questions or concerns, or would like to request an extension. At this stage, your manuscript remains formally under active consideration at our journal; please notify us by email if you do not wish to submit a revision and instead wish to pursue publication elsewhere, so that we may end consideration of the manuscript at PLOS Biology.

When you are ready to submit a revised version of your manuscript, please go to https://www.editorialmanager.com/pbiology/ and log in as an Author. Click the link labelled 'Submissions Needing Revision' where you will find your submission record. 

Best wishes,

Roli

Senior Editor

PLOS Biology

REVIEWERS' COMMENTS:

Reviewer #1:

Summary

Martin et al. present a largely descriptive, but compelling, example of the recent spread of a neo-sex chromosome that is perfectly genetically linked to a male-killing endosymbiont. They show that the spread of these elements in this population has been recent and dramatic. Furthermore, the elements are perfectly linked, emphasizing the impact of sex-biased inheritance in amplifying the effects of a presumed selective sweep. 

Overall the manuscript is extremely well written and the data are clearly presented. I have no major concerns with the methods presented and I am certain this will be of interest to the broad readership of PLOSbio. 

Major

Overall I like the observation and generally think it’s possible that spread of a selfish element has resulted in the spread of genetically linked mtDNA and the neo-W chromosome. However, it is also possible that selection has favored the neo-W either on its own or in addition to the spiroplasm infection. For example, it might be that the neo-sex chromosome experiences some amount of meiotic drive. In fact, if figure S6B reflects the average sex ratio associated with this chromosome, then the female-skew is significantly in excess of the expectation (p = 0.047, binomial test). In either event, the sex ratio of the cured line should be carefully examined and reported as an integral part of this work. 

Furthermore, we might also expect meiotic drive of a linked-sex chromosome to evolve as a consequence of spiroplasm infection. I.e., this would mitigate the cost of male-killing by reducing the base rate of males. In either case, the combination of meiotic drive and female-skewed sex ratios could then also be adaptive for both genetic elements. 

I acknowledge these possibilities are complicated and clearly there is not sufficient data to confidently exclude any (though again, what is the sex ratio of the cured line?), but the text currently presents a simple view that the mtDNA and neo-sex chromosomes are hitchhiking on the spiroplasm infection. A much more nuanced discussion is therefore needed to explain this observation and clarify the possible evolutionary causes. 

Additionally, I understand that the coloration pattern is of historical significance, but I am unclear why so much of the work focuses on this. It seems like an interesting accident that there’s an obvious phenotype associated with this neo-sex chromosome. In either event, I recommend reducing the section that focuses on coloration alleles since it is somewhat aside the main novelty of this work. 

Minor

Lines 135-173. The weak association on chromosome 22 might also reflect errors in the scaffolding. The authors should acknowledge this alternative explanation, but need not discuss in detail since the overall pattern is obvious and this does not impact their primary conclusions. 

Lines 189-191. I find the case in Joron et al. (2011) fairly compelling, though perhaps the authors are leaning on the identification of specific candidate genes. Regardless, I think this statement of novelty is pretty weak and recommend removing it. 

Lines 214-215. Sample size is not mentioned in the main text for the cross experiment and this made me wonder how strong the results are. In figure S6 it is very clear that the sample size is sufficient to confidently support the authors’ statement. I recommend including a short reference to the sample size and a P-value in the main text. 

Line 250. The 100kb windows are non-independent due to linkage. I do not think a Wilcoxon test is the right choice for this. Though that’s probably conservative here since the results support the null. 

LIne 254. “Linage” should be lineage. 

Fig 5. I find the crossing lines in the figure to be slightly confusing as they imply non-concordance between the neo-W and symbiont genealogies, when in fact, there is no evidence for this. I recommend rearranging the tree so these can be displayed cleanly as parallel lines. 

Line 310. “unlinked” should probably be “physically unlinked”. 

The github repository appears to contain a large array of genomic analysis scripts. They are well documented, but it would be good to include the current versioning somewhere in the manuscript in case the scripts are changed at a later time.

Reviewer #2:

Martin et al. have carried out a really interesting study that I think should be published. The authors provide pretty convincing support that a neo-W chromosome has hitchhiked to relatively high frequency via the spread of male-killing spiroplasma. This same (neo-W) chromosomal region contains two colour patterning loci, and due to male-killing the focal population consists of only infected females that carry identical colour patterning alleles. Their analyses provide some candidate genes for this colour pattern variation, although support is limited. Phylogenetic analyses suggest congruence of spiroplasma with the neo-W, supporting hitchhiking of the neo-W with spread of spiroplasma through the population. The lack of female recombination combined with male killing results in a population of only infected females that carry the same colour patterning allele. I have several comments that I hope the authors can address. I have put an asterisk on those comments that I think are the most important, but generally I like this paper and think it contains some really exciting biology that readers will enjoy.

*Line 111: The authors should use dxy and compare to the fst results here since dxy makes interpretation much easier. Also, fst and population size are not related unless I misunderstand.

Line 168: Be more precise here instead of stating “nearly perfect”.

*Lines 174-181: How many other genes are in this area, and how many of those have strongly associated SNPs? There are not many data to really implicate arrow here, and informing the reader of other associated SNPs/genes would be useful. Perhaps a supplemental table for both the yellow and arrow regions that reports SNPs and genes would be useful.

*Line 192: Is obtaining long read (nanopore eg) difficult for some reason in this system? I ask because it would be incredibly useful here, for isolation of the neo W below, etc. In a few weeks it seems the authors could have these data, which would enable them to answer several of the outstanding questions. Unless this is particularly difficult in this system I would urge the authors to consider it.

Line 197: How divergent is H. Melpomene? This would be useful to know when considering the synteny comparison.

*Line 210: Are your species infected with Wolbachia or other endosymbionts? Perhaps this is reported and I missed it, but knowing that spiroplasma is the only reproductive manipulator in the text here is crucial.

Line 227: How is this region “neutral”? Please change the language unless there is compelling support for this.

*Also, do you have data indicating spiroplasma was cured? Tetracycline is often required over multiple generations to ensure low titer infections do not persist. This will also eliminate all of the gut/other microbes? Were any steps taken to reseed the microbiome (eg allowing females time on food where other individuals had previously eaten)? It seems unlikely that anything else is influencing sex ratios, but the authors should provide more detail to convince the reader.

Line 263: What accounts for the 20% reduction? This surprised me and seems much higher than theoretically expected, no?

*Line 285: Based only on reads? Do you have qPCR data? Reads alone are not sufficient in our experience.

*Line 308: B - So the vast majority of nodes have VERY low support? Please report all of the node support and be clear when you can say very little about support for congruence. (I agree with your interpretation, but you should probably be a little more cautious if node support is not great.) 

Line 348: Add a bit more here because it isn’t clear to me why this is support for pre-existing deleterious mutations.

*Line 354: What is the overall distribution of spiroplasma in this species/subspecies, and can you say more about why it seems so restricted here? What is expected in terms of spiroplasma spread? Given the time of the association is it surprising that spiroplasma is so geographically restricted here? Other endosymbionts like Wolbachia rapidly spread over a few decades. Is that not expected for spiroplasma? Hurst, Turelli, or others must have relevant theory on this. It seems like there might be something interesting to say given seasonal fluctuations in immigrant males, which are essential given male killing.

Line 367: How far do males disperse? How does it vary seasonally (specifically).

*Line 459: Is there some justification for the chosen model/approach here? With no partitioning the model assumes everything evolves at the same rate across codon positions. Why not partition the data or assess how assuming rate variation among sites using GTR + G affects the results? 

*Figure 3: The colors in B are difficult at times; specifically, the “x”’s are too light, and distinguishing dorippus and alcippus colors will be difficult for some.

Reviewer #3:

This manuscript recounts a story of a selfish male-killing bacteria driving the evolution of a neo sex chromosome which also carries a tri-allelic colour polymorphism locus. This work goes some way towards working out the genetics and population genetics of this system, using sequence data and a bit of genetics. Truly fascinating stuff. 

The manuscript could use greater clarity and a bit of fleshing out on a few points, however. In particular, it's a very complex system, and I frequently found myself referring back to figure 1, but wishing for more a more comprehensive version of this figure, including information about sample sizes and Spiroplasma infection. A brief overview of the analyses at the beginning of the results would also help.

In general, I found a few comments, e.g., 'To our knowledge, ours is the first example of a butterfly supergene in which the data strongly support the existence of two distinct genes that independently affect colour pattern maintained in LD by suppressed recombination.' strangely defensive. The system is far more interesting that this faint praise would suggest. (Though, as this is purely a stylistic point, I won't complain if they keep this sentence.)

Further, several meiotic drive systems in Drosophila show similar evolutionary patterns (LD between distant loci, most notably in D. pseudoobscura, chromosome-wide hitchhiking), and should be cited where appropriate (see, e.g, Laurracuente et al. 10.1534/genetics.112.141390, Cazemajor GENETICS October 1, 1997 vol. 147 no. 2 635-642, Wu and Beckenbach GENETICS September 1, 1983 vol. 105 no. 1 71-86, Dyer et al. https://doi.org/10.1073/pnas.0605578104). 

There is also a similar kind of story in Heliconius currently on bioarxiv doi: https://doi.org/10.1101/736504. 

Lines 129-137: What are the statistics for the associations mentioned here?

Line 188-- 'distinct functional loci' is vague here

Line 220-- what is the evidence for the complete suppression of recombination in females of this species? (I know its thought to be generally true for Lepidoptera, but thought there were exceptions.)

Line 270-- I found the argument that the selection is due to Spiroplasma vs. colour morphs unconvincing: under some models, recessive alleles can spread. In this case, where the frequency of the recessive allele is elevated by linkage to the W, this seems especially true. 

Figure 3-- having the names repeated over the heterozygotes, with the dominant allele bolded, would help make this figure clearer (particularly when printed in black & white).

Line 346-- can this prediction be tested quantitatively-- is the Pn/Ps ratio for singletons statistically similar to that seen for the high-frequency mutations? It's a bit hard to tell from figure S11, as the colour scale has no numbers, but it seems like it might be a bit higher, suggesting that there has been some accumulation of mutations.

---

## [Decision Letter · Decision Letter 2]

9 Dec 2019

Dear Simon,

Thank you for submitting your revised Research Article entitled "Whole-chromosome hitchhiking driven by a male-killing endosymbiont" for publication in PLOS Biology. I've now obtained advice from the original reviewers and have discussed their comments with the Academic Editor. 

Based on the reviews, we will probably accept this manuscript for publication, assuming that you will modify the manuscript to address the remaining points raised by the reviewers. Please also make sure to address the data and other policy-related requests noted at the end of this email.

IMPORTANT:

a) Please attend to the remaining requests from reviewer #3.

b) Please attend to my Data Policy request further down.

We expect to receive your revised manuscript within two weeks. Your revisions should address the specific points made by each reviewer. In addition to the remaining revisions and before we will be able to formally accept your manuscript and consider it "in press", we also need to ensure that your article conforms to our guidelines. A member of our team will be in touch shortly with a set of requests. As we can't proceed until these requirements are met, your swift response will help prevent delays to publication.

*Copyediting*

*Published Peer Review History*

*Early Version*

*Submitting Your Revision*

Sincerely,

Roli

Senior Editor

PLOS Biology

ETHICS STATEMENT:

The Ethics Statements in the submission form and Methods section of your manuscript should match verbatim. Please ensure that any changes are made to both versions.

-- Please include the full name of the IACUC/ethics committee that reviewed and approved the animal care and use protocol/permit/project license. Please also include an approval number.

-- Please include the specific national or international regulations/guidelines to which your animal care and use protocol adhered. Please note that institutional or accreditation organization guidelines (such as AAALAC) do not meet this requirement.

-- Please include information about the form of consent (written/oral) given for research involving human participants. All research involving human participants must have been approved by the authors' Institutional Review Board (IRB) or an equivalent committee, and all clinical investigation must have been conducted according to the principles expressed in the Declaration of Helsinki.

DATA POLICY:

Regardless of the method selected, please ensure that you provide the individual numerical values that underlie the summary data displayed in the figure panels. My understanding is that you have deposited these in Dryad, but we will need a reviewer link or login to check the Dryad data provision before we can proceed. NOTE: the numerical data provided should include all replicates AND the way in which the plotted mean and errors were derived (it should not present only the mean/average values).

Please also ensure that figure legends in your manuscript include information on where the underlying data can be found (i.e. Dryad URL), and ensure your supplemental data file/s has a legend.

REVIEWERS' COMMENTS:

Reviewer #1:

The authors have completely addressed my concerns.

Reviewer #2:

The authors have provided thoughtful responses to each of my comments. I have no additional comments or requests. I look forward to seeing this very interesting work published.

Reviewer #3:

I find this manuscript, already very interesting, has been further improved by the changes made in response to the reviews, in particular the addition of more statistical analyses of the arguments made, and further discussion of points that were a bit neglected before. I think there are a few niggling points where additional clarity would make the manuscript even better, but I'm sure the authors can easily address these suggestions.

-Forgive me if I'm being dense, but I'm struggling to understand the meiotic drive argument. As I understand it, the broods of neo-W carrying females are very female biased due to the strong association between the neo-W and the male-killing Spiroplasma (perfect, in the case of the sample in this study). I think the other reviewer perhaps has some Fisherian sex ratio argument in mind, but are they assuming that the neo-W also occurs in non-Spiroplasma infected females (thus reducing the 'base rate' of males)? In any case, I think it would help to flesh out how this system would work for readers (like me) who don't find it intuitive. I do agree that with the larger point of reviewer 1, however, that the added nuance in the discussion has improved the paper, as has the discussion of the sex ratio of cured lines. 

-What is the association between the neo-W and Spiroplasma in the wild? (Or, perhaps there's more data for male-killing and the colour polymorphism.) Ten wild-caught females with the right colour pattern and no male killing are document in reference 48-- do we know what proportion this is of the population? Presumably Spiroplasma isn't inherited perfectly, so there should be some neo-W females with no male-killing. If the central hypothesis of the paper is right, however, these should be rare.

Line 271-- three orders of magnitude?

Line 338-- 'this' is ambiguous here; can this sentence be rephrased?

Line 380-381-- 'higher than for singletons on autosomes' may require a little more explanation. Something like 'higher than expected number of mutations captured on a random copy of an autosome', but more gracefully phrased, would be good.

There are random extra spaces throughout.

---

## [Editor Report · Decision Letter 3]

23 Jan 2020

Dear Dr Martin,

On behalf of my colleagues and the Academic Editor, Dr. Harmit S. Malik, I am pleased to inform you that we will be delighted to publish your Short Reports in PLOS Biology. 

Early Version

PRESS 

Kind regards,

Krystal Farmer,

Development Editor 

PLOS Biology

on behalf of

Roland Roberts,

Senior Editor

PLOS Biology